# The cytoplasmic synthesis and coupled membrane translocation of eukaryotic poly-phosphate by signal-activated VTC complex

Zeyuan Guan[1,7], Juan Chen[1,7], Ruiwen Liu[1,7], Yanke Chen[2,7], Qiong Xing [3,7], Zhangmeng Du[1], Meng Cheng[1], Jianjian Hu[1], Wenhui Zhang[1], Wencong Mei[1], Beijing Wan[1], Qiang Wang [1], Jie Zhang[1], Peng Cheng[1], Huanyu Cai[4], Jianbo Cao[5], Delin Zhang [1], Junjie Yan[1], Ping Yin [1], Michael Hothorn [6] & Zhu Liu [1] ✉

Inorganic polyphosphate (polyP) is an ancient energy metabolite and phosphate store that occurs ubiquitously in all organisms. The vacuolar transporter chaperone (VTC) complex integrates cytosolic polyP synthesis from ATP and polyP membrane translocation into the vacuolar lumen. In yeast and in other eukaryotes, polyP synthesis is regulated by inositol pyrophosphate (PP-InsP) nutrient messengers, directly sensed by the VTC complex. Here, we report the cryo-electron microscopy structure of signal-activated VTC complex at 3.0 Å resolution. Baker's yeast VTC subunits Vtc1, Vtc3, and Vtc4 assemble into a 3:1:1 complex. Fifteen trans-membrane helices form a novel membrane channel enabling the transport of newly synthesized polyP into the vacuolar lumen. PP-InsP binding orients the catalytic polymerase domain at the entrance of the trans-membrane channel, both activating the enzyme and coupling polyP synthesis and membrane translocation. Together with biochemical and cellular studies, our work provides mechanistic insights into the biogenesis of an ancient energy metabolite.

Inorganic polyphosphate (polyP), a molecular fossil that was first described more than 100 years ago, is one of the most ancient, conserved, and enigmatic biomolecules during species evolution, and is abundant (reaching millimolar concentrations) in pro- and eukaryotic cells[1–4]. PolyP is an energy polymer of tens to hundreds of orthophosphate ($PO_4^{3-}$) units linked by high-energy phosphoanhydride bonds. Eukaryotic polyPs mainly accumulate as granules in specific vacuoles called acidocalcisomes, that also store calcium and other divalent cations[5]. PolyP synthesis and accumulation impacts a myriad of cellular functions by taking diverse biological roles ranging from energy storage to cell signaling[6–8]. Regulated metabolism of polyP maintains the

cellular homeostasis of energy and phosphate (Pi)[9–14]. Low levels of polyP impact mitochondrial metabolism[15], cell apoptosis[16], bone mineralization[17], procoagulant and proinflammatory responses[18], and cause defects in mTOR signaling[19]. PolyP controls intracellular divalent cation availability by serving as a chelator[1], protects protein against stress-induced aggregation by functioning as a chaperone[20], covalently modifies proteins and regulates their functions[21], activates ribosomal protein degradation in response to amino acid starvation[22], and mitigates amyloidogenic processes in neurodegenerative diseases[23,24].

The vacuolar transporter chaperone (VTC) complex is the only known polyP polymerase in eukaryotic cells[12,25,26]. In yeast, the

[1]National Key Laboratory of Crop Genetic Improvement, Hubei Hongshan Laboratory, Huazhong Agricultural University, Wuhan 430070, China. [2]Wuhan Institute of Physics and Mathematics, Innovation Academy for Precision Measurement Science and Technology, Chinese Academy of Sciences, Wuhan 430071, China. [3]State Key Laboratory of Biocatalysis and Enzyme Engineering, School of Life Sciences, Hubei University, Wuhan 430062, China. [4]College of Science, Huazhong Agricultural University, Wuhan 430070, China. [5]Public Laboratory of Electron Microscopy, Huazhong Agricultural University, Wuhan 430070, China. [6]Structural Plant Biology Laboratory, Department of Plant Scienes, University of Geneva, Geneva 1211, Switzerland. [7]These authors contributed equally: Zeyuan Guan, Juan Chen, Ruiwen Liu, Yanke Chen, Qiong Xing. ✉e-mail: liuzhu@hzau.edu.cn

vacuolar membrane-localized VTC complex senses inositol pyrophosphate (PP-InsP) nutrient messengers[27] (which accumulate in Pi sufficient growth conditions) in response to changing nutrient environments[9], to stimulate cytosolic ployP synthesis from adenosine triphosphate (ATP)[12], polyP membrane translocation and storage in the vacuolar lumen[12,28] (Supplementary Fig. 1a). Baker's yeast harbors two VTC sub-complexes, containing Vtc1/Vtc2/Vtc4 or Vtc1/Vtc3/Vtc4[12,29,30]. Subunit Vtc1 harbors a trans-membrane (TM) domain, Vtc4, Vtc3, and Vtc2 contain additional cytoplasmic SPX (SYG1/Pho81/XPR1) and TTM (triphosphate tunnel metalloenzyme) domains[31] (Supplementary Fig. 1b). The TTM domain[32] of Vtc4 (TTM$^{Vtc4}$) synthesizes polyP, by transferring the γ-phosphate of ATP onto growing polyP chain[12]. The SPX domains are receptors for cytosolic PP-InsP, and PP-InsP binding stimulates VTC-catalyzed polyP synthesis[9,33]. The TM domains in VTC complex are expected to assemble into a trans-membrane channel for the transport of growing polyP across the vacuolar membrane[12,28,34]. Overall, the VTC complex is a unique membrane protein machinery that integrates functions of a PP-InsP receptor, a polyP polymerase and coupled membrane translocase to generate and store polyP, yet the underlying molecular mechanism remains structurally uncharacterized. To uncover how the VTC complex is assembled and how it is activated to integrate polyP synthesis and membrane translocation, we used cryo-electron microscopy (cryo-EM), together with biochemical and cellular studies by jointly performing in-cell NMR and smFRET analyses, to provide mechanistic insights into the biogenesis of this ancient energy metabolite.

## Results

### Function characterization and structure determination

Due to acidic environment of yeast vacuole, the vacuolar polyP store exhibits specific $^{31}$P chemical shift values, and thus polyP concentrations can be quantified by in-cell $^{31}$P-NMR measurements[35]. In this study, we first performed in-cell $^{31}$P-NMR measurements of polyP in yeast strains to characterize the contribution of each Vtc subunit to vacuolar polyP synthesis (Fig. 1). We found that *vtc1* and *vtc4* are essential for the generation of the vacuolar polyP, as previously described[29]. Deletion of either *vtc2* or *vtc3* partially decreased polyP levels, whereas *vtc2/vtc3* double mutants contained no detectable polyP. These data collectively suggest that both the Vtc1/Vtc2/Vtc4 and the Vtc1/Vtc3/Vtc4 sub-complexes can generate vacuolar polyP, consistent with previous findings[12,29,30]. Here we focus our studies on the Vtc1/Vtc3/Vtc4 complex.

We reconstituted a Vtc1/Vtc3/Vtc4 complex that harbors polyP polymerase activity and that synthesizes polyP in an InsP$_6$ dose-dependent manner (Supplementary Fig. 2a-d, and Methods), consistent with previous findings that InsPs and PP-InsPs can activate the VTC polymerase activity in vivo[9,33,36]. To assess whether the VTC-synthesized polyP in vitro associates with the entire trans-membrane complex, we incubated VTC complex with ATP and InsP$_6$ and

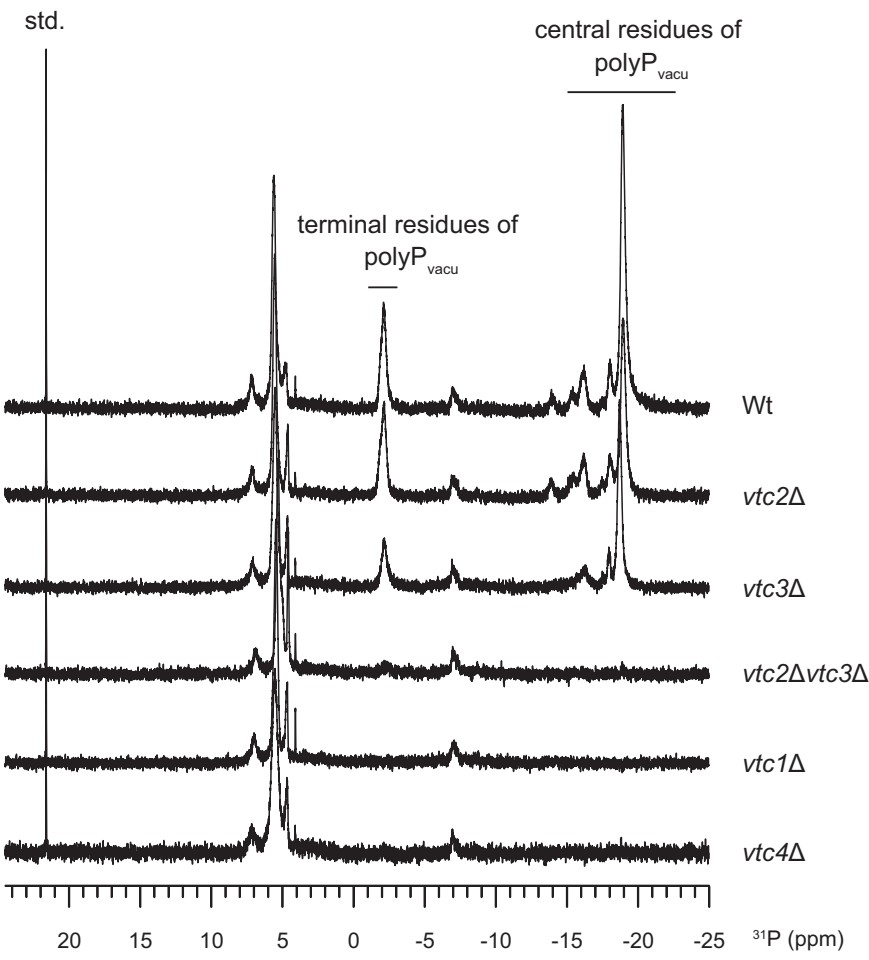

**Fig. 1 | In-cell $^{31}$P-NMR measurements of vacuolar polyP in yeast strains.** Chemical shifts of the terminal and central residues of vacuolar polyP are indicated, respectively. MDP was used as the std. reference. Wt indicates the wild-type BY4741 yeast strain. *vtc2Δ, vtc3Δ, vtc2Δvtc3Δ, vtc1Δ, vtc4Δ* indicates BY4741 yeast strain with deletion of *vtc2, vtc3, vtc2* and *vtc3, vtc1, vtc4* gene, respectively.

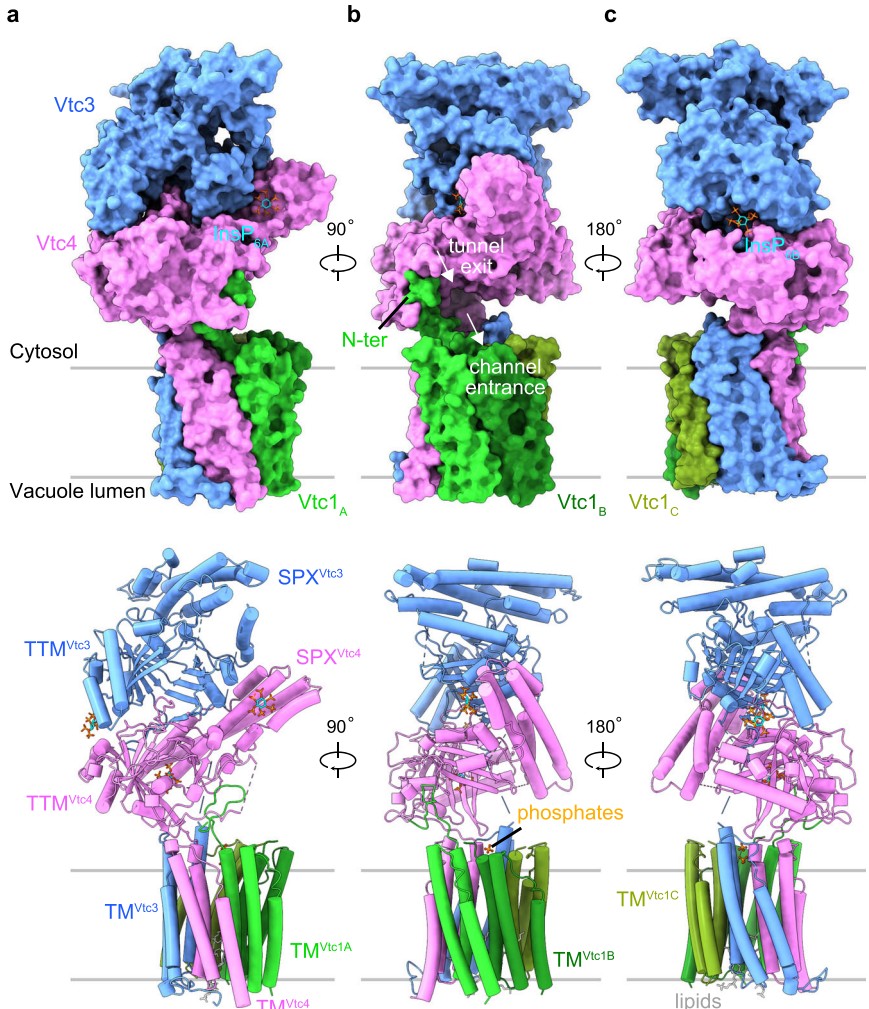

**Fig. 2 | Structure of the InsP6-activated VTC complex. a–c** Overall structure of Vtc1/Vtc3/Vtc4$_{R264A/R266A/E426N}$ in complex with InsP6. Surface (*Top*) and cartoon (*Bottom*) representations are displayed in the same perspective. Vtc3 and Vtc4 are colored in blue and pink, respectively. The protomer A, B, and C of Vtc1 are colored separately in gradient green. Molecules of InsP6, phosphate and lipid are shown in stick representations. The exit tunnel of catalytic TTM$^{Vtc4}$ and the entrance channel of the trans-membrane region are marked with white arrows.

subjected the reaction mixture to size-exclusion chromatography (SEC) (Supplementary Fig. 2e). We isolated the VTC-containing elution fractions and probed for the presence of polyP by DAPI[37] (4′,6-diamidino-2-phenylindole) staining. We found long polyP chains associated with our reconstituted VTC complex (Supplementary Fig. 2e, f), indicating that the produced polyP may be transited into the trans-membrane channel.

We next inactivated VTC by substituting three conserved amino acids in the TTM domain of the catalytic Vtc4 subunit[12] and prepared a catalytically impaired Vtc1/Vtc3/Vtc4$_{R264A/R266A/E426N}$ complex in the presence of 1 mM InsP6 (a commercially available surrogate for the bioactive PP-InsPs[9,38]) (Supplementary Fig. 3). For this sample, we could collect high-quality cryo-EM micrographs for single particle analysis. A 3D reconstructed density map of this InsP6-activated VTC complex was refined at an overall resolution of 3.0 Å (Supplementary Fig. 4). The high-quality map allowed us to build atomic models for all domains of each Vtc subunits, including SPX domains of Vtc3 and Vtc4, TTM domains of Vtc3 and Vtc4, and TM domains of Vtc3, Vtc4 and Vtc1 (Fig. 2, Supplementary Figs. 4, 5, and Supplementary Table 1).

### Overall structure of InsPs-activated VTC complex

The VTC complex structure assembles into a pentamer with Vtc1, Vtc3 and Vtc4 in a stoichiometry ratio of 3:1:1 (Fig. 2). The TM domains of

VTC complex have been predicted to contain at least nine transmembrane helices[28,39,40]. Our cryo-EM structure reveals that the TM region contains fifteen trans-membrane helices, wherein the TM domain of each subunit comprises three trans-membrane helices (Fig. 2 and Supplementary Fig. 5b–f). The SPX domain of Vtc3 (SPX$^{Vtc3}$) mainly associates with TTM$^{Vtc3}$ at the membrane-distal region of VTC complex. The SPX$^{Vtc4}$ and TTM$^{Vtc4}$ domains are sandwiched between TTM$^{Vtc3}$ and the TM domains, with SPX$^{Vtc4}$ tightly interacting with TTM$^{Vtc4}$ (Fig. 2a). The exit tunnel of the catalytic TTM$^{Vtc4}$ domain is positioned at the entrance of the trans-membrane channel (Fig. 2b and Supplementary Fig. 6a). TTM$^{Vtc4}$ and TM$^{Vtc4}$ are connected by a well-defined loop (residues 472–618, named TTM$^{Vtc4}$-TM$^{Vtc4}$ connection hereafter), and this loop interacts with the N-terminus of one Vtc1 protomer (Vtc1$_A$, including residues 1–21) at the membrane-proximal region (Fig. 2b and Supplementary Fig. 6a). No well-defined EM density could be located for the TTM$^{Vtc3}$-TM$^{Vtc3}$ connection, the N-termini of the other two Vtc1 protomers (Vtc1$_B$ and Vtc1$_C$) appear disordered, probably owing to structural flexibility.

Three InsP6 molecules were identified in the VTC complex structure (Supplementary Fig. 5g). InsP$_{6A}$ binds to the basic surface patch of the PP-InsP sensing SPX$^{Vtc4}$ domain (Fig. 2a), in a conformation that slightly differs from the position of InsP6 previously observed in an isolated SPX$^{Vtc4}$ crystal structure[9] (Supplementary

Fig. 6b). In line with this, conformational differences of InsP$_6$ binding to other fungal and plant SPX domains have been previously reported[9,38,41]. A second InsP$_{6B}$ molecule binds simultaneously to TTM$^{Vtc3}$ and TTM$^{Vtc4}$, likely promoting the association of the two tunnel domains (Fig. 2c and Supplementary Fig. 6c). A third InsP$_{6C}$ appears in the positively charged tunnel domain of TTM$^{Vtc4}$ (Supplementary Fig. 6d), in a binding site that is normally occupied by the ATP substrate. The unspecific binding of other, negatively charged ligands such as sulfate, orthophosphate, pyrophosphate and polyphosphate to VTC TTM domains has been previously observed[12]. This feature of the TTM$^{Vtc4}$ tunnel enables the polymerase domain to initially bind pyrophosphate as a primer to start the synthesis of polyP chain[12]. Two phosphates ions and two lipid molecules bind to the cytoplasmic and luminal side of the VTC transmembrane channel, respectively (Fig. 2b, c and Supplementary Figs. 5h and 6e).

### PP-InsP messenger sensing and polymerase activation

SPX domains are eukaryotic PP-InsP receptors, in which basic residues termed phosphate binding cluster (PBC) and lysine surface cluster (KSC) together provide a binding surface for PP-InsPs[9,38,41]. In the structure of our InsP$_6$-activated VTC complex, a molecule of InsP$_6$ (InsP$_{6A}$) is bound to SPX$^{Vtc4}$ receptor. No InsP$_6$ density is observed in SPX$^{Vtc3}$, possibly because the PBC (Y22, K26, K130) and KSC (K126, K129, K133) surfaces are partially buried and thus inaccessible for InsP$_6$/PP-InsP binding (Supplementary Fig. 8). InsP$_{6A}$, is mainly recognized by the SPX$^{Vtc4}$ PBC (Y22, K26, K133) and KSC (K129, K132, K136) residues (Fig. 3a), as previously reported[9]. Previous findings revealed that point mutations targeting their KSC or PBC residues in SPX domains reduced InsP$_6$ and PP-InsP binding[9]. We generated these point mutations in the full VTC complex and assessed their impact on InsP$_6$-stimulated polyP synthesis (Fig. 3b, c and Supplementary Fig. 9). We found that mutation of SPX$^{Vtc4}$ residues in either PBC (Vtc4$^{SPX,PBC}$, Vtc1/Vtc3/Vtc4$_{Y22F/K26A/K133A}$) or KSC (Vtc4$^{SPX,KSC}$, Vtc1/Vtc3/Vtc4$_{K129A/K132A/K136A}$) both reduced InsP$_6$-stimulated polyP synthesis (Fig. 3b, c). This is in line with a previous report, in which SPX$^{Vtc4}$ point mutations showed reduced polyP production in intact yeast vacuoles[9]. In contrast, mutation of SPX$^{Vtc3}$ PBC (Vtc3$^{SPX,PBC}$, Vtc1/Vtc3$_{Y22F/K26A/K130A}$/Vtc4) or KSC (Vtc3$^{SPX,KSC}$, Vtc1/Vtc3$_{K126A/K129A/K133A}$Vtc4) residues had little effect on polyP synthesis (Fig. 3b, c), in agreement with our complex structure. Together, these findings reveal that it is the SPX domain of Vtc4, rather than that of Vtc3, which acts as the PP-InsP nutrient sensor in fully assembled VTC complex.

In the VTC complex structure, an additional InsP$_6$ molecule (InsP$_{6B}$) is bound to a positively charged cleft that is formed by TTM$^{Vtc4}$ and TTM$^{Vtc3}$ (Fig. 2c and Supplementary Figs. 6c and 10a). This binding cleft contains K300, R302 and K320 of TTM$^{Vtc4}$, and K333, K362, K364 and Y380 of TTM$^{Vtc3}$. We hypothesized that InsP$_{6B}$ acts as a molecular glue promoting the association of TTM$^{Vtc4}$ and TTM$^{Vtc3}$ in the complex. In line with this, we found that the isolated TTM$_{189-480}^{Vtc4}$ (including residues 189–480) and TTM$_{183-553}^{Vtc3}$ (including residues 183–553) domains can interact in size-exclusion chromatography (SEC) experiments only in the presence of InsP$_6$, whereas they eluted as isolate peaks in the absence of InsP$_6$ (Supplementary Fig. 10b). Point mutations targeting the cleft residues in TTM$_{189-480}^{Vtc4}$ (TTM$_{189-480}^{Vtc4,mutant}$, K300A/R302A/K320A) or TTM$_{183-553}^{Vtc3}$ (TTM$_{183-553}^{Vtc3,mutant}$, K333A/K362A/K364A/Y380F) disrupted this InsP$_6$-promoted TTM$_{189-480}^{Vtc4}$ – TTM$_{183-553}^{Vtc3}$ association (Supplementary Fig. 10b). However, cleft mutations in either TTM$^{Vtc4}$ (Vtc4$^{TTM,mutant}$, Vtc1/Vtc3/Vtc4$_{K300A/R302A/K320A}$) or TTM$^{Vtc3}$ (Vtc3$^{TTM,mutant}$, Vtc1/Vtc3$_{K333A/K362A/K364A/Y380F}$/Vtc4) had little impact on InsP$_6$-stimulated polyP synthesis by VTC complex (Supplementary Fig. 10c). The possible physiological role of our observed InsP$_6$-promoted TTM$^{Vtc4}$-TTM$^{Vtc3}$ interaction needs further exploration.

### Mechanistic coupling of polyP synthesis and membrane translocation

In the InsP$_6$-activated VTC complex, the catalytic TTM$^{Vtc4}$ tunnel is located close to the TM region (Supplementary Fig. 6a). The inner TM helix 1 (TM1) of each subunit TM domain (five TM1s in total) form the membrane channel (Supplementary Fig. 6e). The cytoplasmic entrance of this channel opens to the polyP exit tunnel of the TTM$^{Vtc4}$ enzyme (Supplementary Fig. 6a), and is occupied by two phosphate ions (Supplementary Fig. 6e). The TTM$^{Vtc4}$-TM region provides a continuous and positively charged belt-like path from the enzyme tunnel exit to the TM channel entrance, likely facilitating the transport of the nascent polyP chain towards the pore (Fig. 4a). Superimposing the previously reported structure of isolated TTM$^{Vtc4}$ domain bound to polyP[12] with TTM$^{Vtc4}$ in our VTC complex structure, we find the synthesized polyP to follow the positively charged belt towards the translocation channel (Fig. 4a), providing a mechanistic rationale for the coupled and concomitant synthesis and membrane translocation of polyP[12,28]. The two phosphate ions in the translocation channel map below the synthesized polyP chain (Fig. 4a), suggesting that they could mimic a polyP molecule entering the pore.

The isolated Vtc4$^{TTM}$ domain is a slow enzyme for polyP synthesis, whereas VTC is a very efficient enzyme in vivo[12]. Consequently, a coupled model for polyP synthesis and membrane translocation was proposed[12,28], in which a driving force from the TM region pulls the nascent polyP chain into the TM channel for translocation, thereby discharging the negatively charged polymer from the positively charged catalytic center and allowing the addition of new phosphate residues. Our complex structure reveals that the interior wall of the upper translocation channel is positively charged by a series of basic residues in TM1s, including K24 and R31 of TM1$^{Vtc1}$, K694, K698, R705, and R709 of TM1$^{Vtc3}$, and K622 and R629 of TM1$^{Vtc4}$ (Fig. 4b and Supplementary Fig. 11). The narrowest position of the channel is surrounded by R31 of TM1$^{Vtc1}$, R709 of TM1$^{Vtc3}$, and R629 of TM1$^{Vtc4}$, with a radius of 1.5 Å (Supplementary Fig. 11). We speculate that this positive potential could be the driving force for polyP catalysis-discharge and transmembrane-translocation. We thus replaced these basic residues in the VTC complex with alanine. We found that mutations in TM1$^{Vtc1}$, TM1$^{Vtc3}$, or TM1$^{Vtc4}$ had various effects on polyP synthesis in vitro (Fig. 5a). Mutation of K24 and R31 to alanine in TM1$^{Vtc1}$ (Vtc1$^{TM1,mutant}$, Vtc1$_{K24A/R31A}$/Vtc3/Vtc4) increased VTC activity (Fig. 5a). Mutation of several basic residues in TM1$^{Vtc3}$ (Vtc3$^{TM1,mutant}$, Vtc1/Vtc3$_{K694A/K698A/R705A/R709A}$/Vtc4) or TM1$^{Vtc4}$ (Vtc4$^{TM1,mutant}$, Vtc1/Vtc3/Vtc4$_{K622A/R629A}$) reduced polyP synthesis in vitro (Fig. 5a), indicating a reduced catalysis-discharge from the TTM$^{Vtc4}$ domain, possibly caused by a weakened driving force from the translocation channel. Based on our mutational analyses, the TM1 domain of Vtc3 appears to play the dominant role in polyP discharging. Furthermore, in-cell $^{31}$P-NMR measurements showed that mutations targeting on above-mentioned channel residues in any subunit eliminated the VTC-generated vacuolar polyP store in yeast strains (Fig. 5b), suggesting that these positively charged residues are necessary for the translocation of the negatively charged polyP chain across the vacuolar membrane. Together, our structure-guided functional analyses rationalize previous biochemical and genetic findings that the VTC couples cytosolic polyP synthesis with efficient, concomitant translocation of the nascent chain into the vacuole[28].

### PP-InsP binding orients the polymerase domain at the entrance of the trans-membrane channel

Our complex structure further indicates that the coupling of polyP synthesis and translocation could be facilitated by the exact orientation of the TTM$^{Vtc4}$ catalytic module on top of the polyP translocation channel (Fig. 5c and Supplementary Fig. 6a). The orientation of TTM$^{Vtc4}$ is mainly stabilized by the interaction between the TTM$^{Vtc4}$-TM$^{Vtc4}$ connection and the N-terminus of Vtc1$_A$. Deleting the

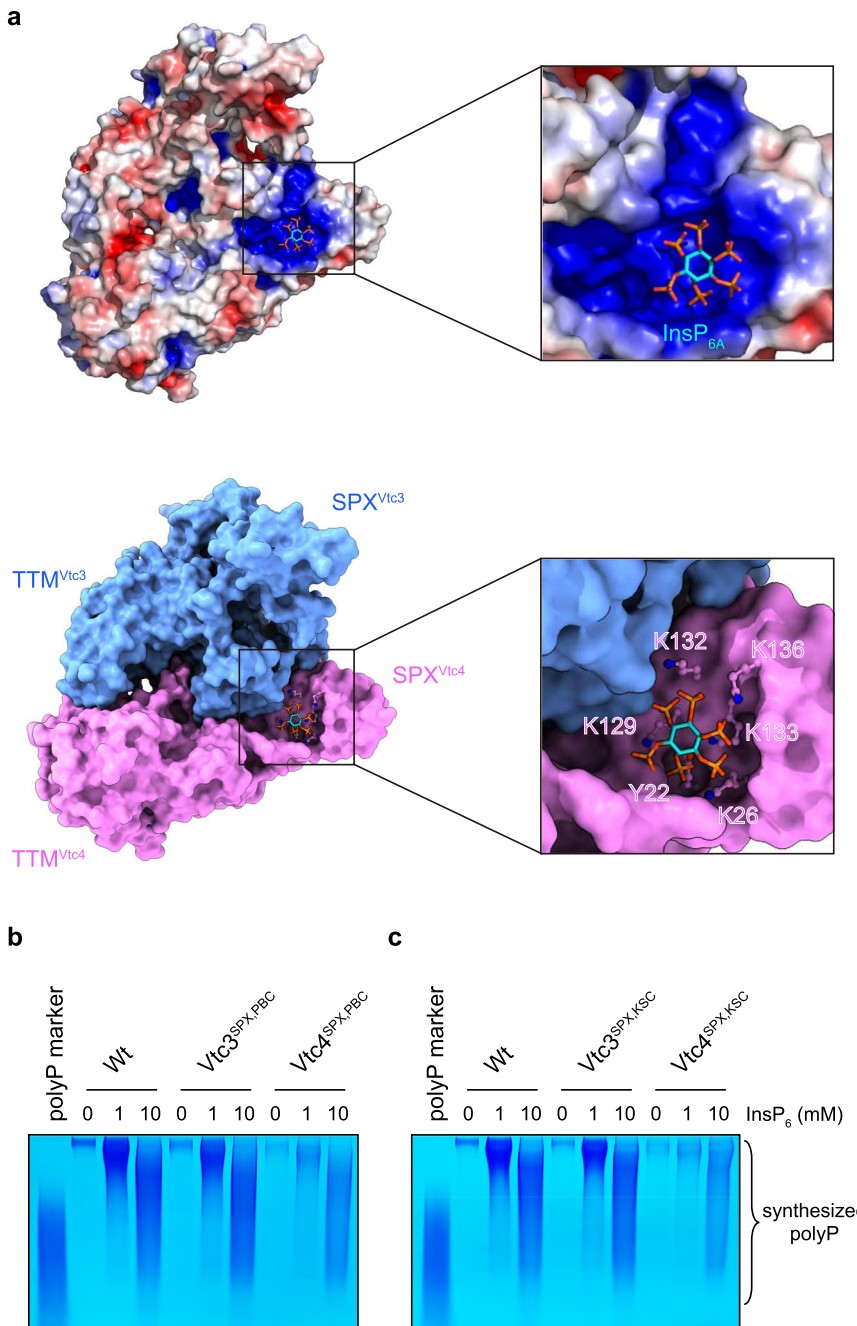

**Fig. 3 | Activation of the VTC complex by SPX$^{Vtc4}$-sensed InsP$_6$ messenger.**
**a** Electrostatic surface (*Top*) and molecular surface (*Bottom*) representation of SPX and TTM domains in InsP$_6$-activated Vtc1/Vtc3/Vtc4$_{R264A/R266A/E426N}$ complex, displayed in the same perspective. Electrostatic potential is calculated with the APBS plugin in PyMOL, colored and displayed in a scale from red (−5 kT/e) to blue (+5 kT/e). SPX$^{Vtc4}$ PBC (Y22, K26, K133) and KSC (K129, K132, K136) residues are shown in stick representation. **b**, **c** Assay of VTC-catalyzed polyP synthesis. 5 μM protein complex were used and reactions were performed at required InsP$_6$ concentrations for 4 h. A commercial polyP with an average chain length of 60 residues was used as the marker. Wt, Vtc1/Vtc3/Vtc4; Vtc3$^{SPX,PBC}$, Vtc1/Vtc3$_{Y22F/K26A/K130A}$/Vtc4; Vtc4$^{SPX,PBC}$, Vtc1/Vtc3/Vtc4$_{Y22F/K26A/K133A}$; Vtc3$^{SPX,KSC}$, Vtc1/Vtc3$_{K126A/K129A/K133A}$Vtc4; Vtc4$^{SPX,KSC}$ Vtc1/Vtc3/Vtc4$_{K129A/K132A/K136A}$. Source data are provided as a Source Data file. The experiments were repeated three times independently with similar results.

N-terminus of Vtc1 (Vtc1$^{ΔN}$, Vtc1$_{Δ1−21}$/Vtc3/Vtc4) decreased the catalytic activity of VTC (Fig. 5a), and abolished the generation of vacuolar polyP in the mutant yeast strains (Fig. 5b). We therefore speculate that the Vtc1 N-terminus deletion may disrupt the coupling of polyP synthesis to its membrane translocation, by releasing the TTM$^{Vtc4}$-TM$^{Vtc4}$ connection and resulting in TTM$^{Vtc4}$ to move away from the translocation channel, simultaneously hampering polyP synthesis and membrane translocation.

To clarify if deletion of the Vtc1 N-terminus may result in the envisioned movement of the TTM$^{Vtc4}$ catalytic domain, we performed single-molecule fluorescence resonance energy transfer (smFRET) analysis, a method to characterize protein dynamics and conformational changes at single-molecule level in solution[42–44]. We specifically labeled the reconstituted Vtc1/Vtc3/Vtc4 complex with fluorescent probes of Alexa488 (FRET donor) and Cy5 (FRET acceptor) though thiol-maleimide chemical conjugation, by substituting all intrinsic

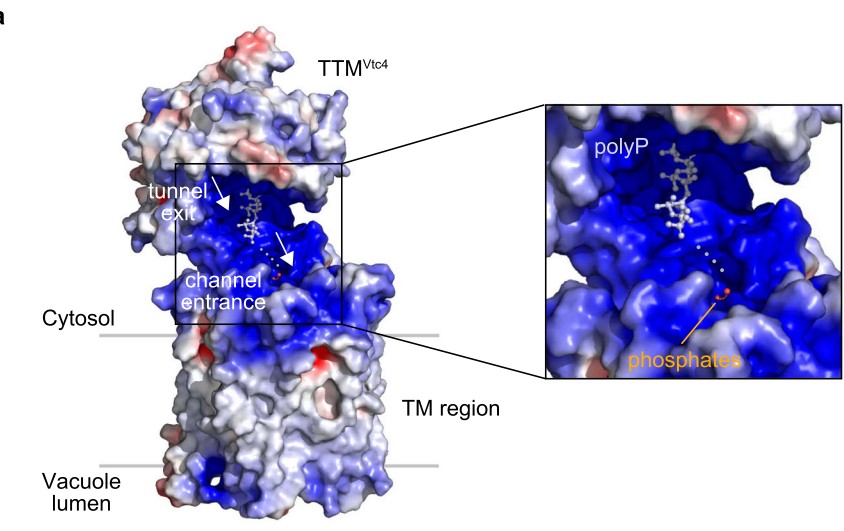

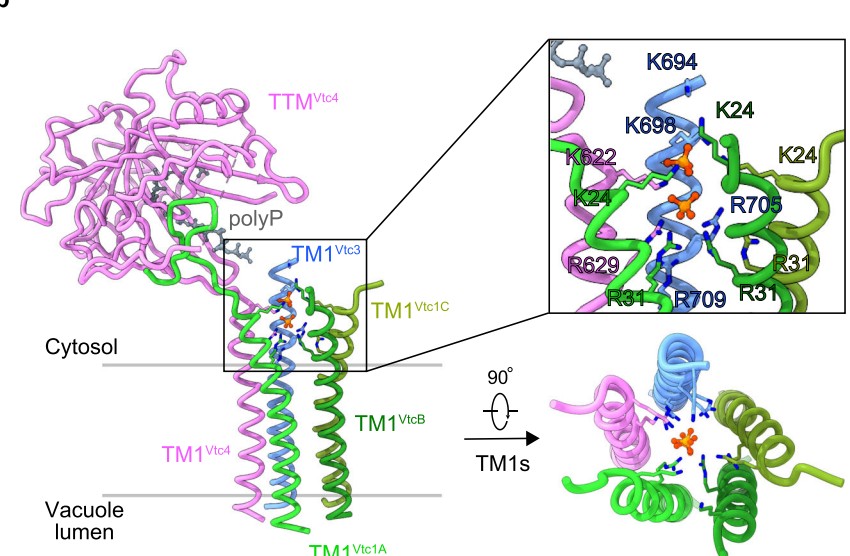

**Fig. 4 | A continuous and positively charged belt on the path from the tunnel exit of catalytic TTM$^{Vtc4}$ to the entrance of trans-membrane channel.**
**a** Electrostatic surface of TTM$^{Vtc4}$ and TM region in InsP$_6$-activated Vtc1/Vtc3/Vtc4$_{R264A/R266A/E426N}$ complex. It is calculated with the APBS plugin in PyMOL, colored in terms of electrostatic potential, and displayed in a scale from red ($-5$ kT/e) to blue ($+5$ kT/e). The previously reported structure of isolated TTM$^{Vtc4}$ domain bound to polyP (3G3Q.PDB[12]) is superimposed to TTM$^{Vtc4}$ in the entire VTC complex, with an RMSD of 0.62 Å. PolyP in the previously reported structure and phosphates in the entire VTC complex are shown in stick representation. Gray dashed line indicates a potential pathway for the transmission of nascent polyP chain from synthesis to translocation. **b** Representation of the TTM$^{Vtc4}$ and TM1s-assembled trans-membrane channel. The previously reported isolated TTM$^{Vtc4}$ domain in complex with polyP (3G3Q.PDB) is also superimposed as in (**a**), and only polyP is represented for clarity. Basic residues of the five TM1s in the interior wall of upper translocation channel and the occupied phosphates are zoomed-in and shown in stick representations. Top view of this channel is also represented.

cysteine in VTC complex and further generating additional cysteine to the solvent-exposed K415 and K689 in the TTM$^{Vtc4}$ and TM$^{Vtc4}$, respectively (Fig. 5c and Supplementary Fig. 12a). Probes-conjugated VTC complex exhibits polyP-synthesizing capacity comparable to wild-type VTC complex, and also in an InsP$_6$ dose-dependent manner (Supplementary Fig. 12b). These results indicate that our protein engineering does not perturb VTC structure and function. By performing smFRET analysis, we found that for the intact VTC complex (Wt), InsP$_6$ presence changed the smFRET profile from a broadened and low-FRET distribution (Fig. 5c, gray profile, corresponding to the inactive conformation) to a sharp and more populated high-FRET species (Fig. 5c, black profile, corresponding to the activated conformation). This change of smFRET distribution indicates that InsP$_6$ binding closes TTM$^{Vtc4}$ to TM region in the VTC complex. Based on our InsP$_6$-activated VTC complex structure, we modeled the fluorescent probes at their labeling sites and calculated the probe-probe distance. The calculated average distance is $42.6 \pm 6.5$ Å between geometric centers of the two chromophores, expecting a theoretical FRET efficiency of 0.77 ($R_0 = 52$ Å[45]) (Supplementary Fig. 12c). This value is consistent with the observed efficiency of the InsP$_6$-enriched high-FRET species (Fig. 5c, black profile), corroborating our InsP$_6$-activated VTC structure. Furthermore, when we deleted the Vtc1 N-terminus (Vtc1$^{\Delta N}$), regardless of the presence of InsP$_6$, the smFRET profile distributed similar to that of intact VTC complex in the absence of InsP$_6$ (Fig. 5c). These results indicate that the Vtc1 N-terminus deletion causes TTM$^{Vtc4}$ to move away from TM$^{Vtc4}$ in the TM region. As a result, deletion of the Vtc1 N-terminus results in a mutant VTC complex that can no longer be activated by the InsP$_6$ stimulus (Fig. 5a, b). Together, the N-terminus of Vtc1 is crucial for VTC activation, by establishing the TTM$^{Vtc4}$–TM region interaction upon InsP$_6$/PP-InsP sensing, to orient

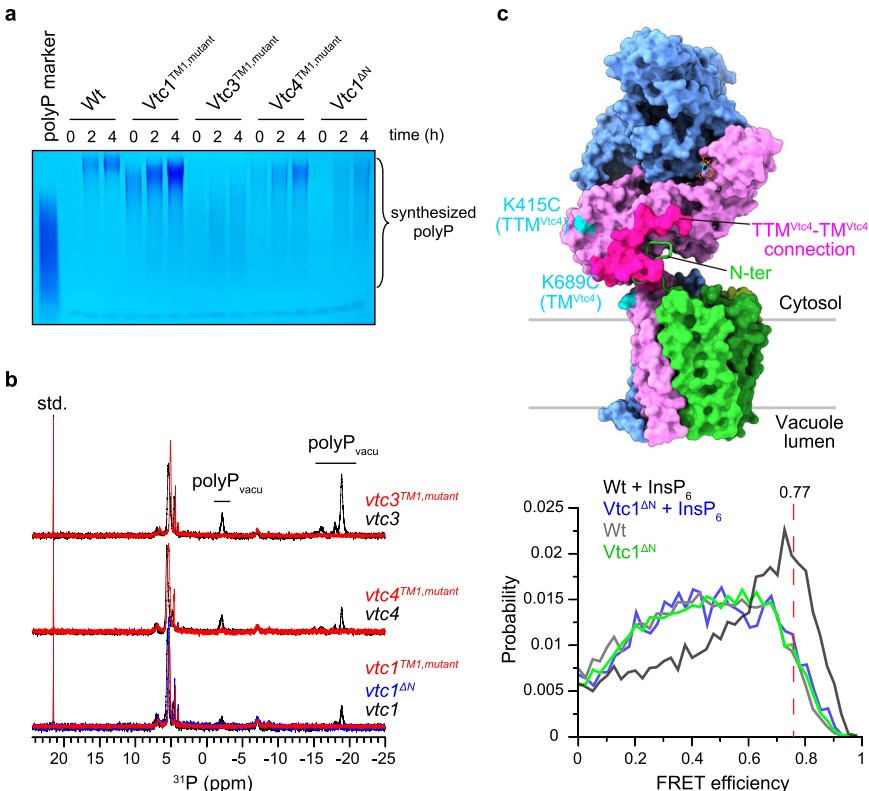

**Fig. 5 | Mechanistic coupling of polyP synthesis and membrane translocation.** **a** Assay of VTC-catalyzed polyP synthesis. 5 µM protein complex were used and reactions were performed with required times in the presence of 1 mM InsP$_6$. Wt, Vtc1/Vtc3/Vtc4; Vtc1$^{TM1,mutant}$, Vtc1$_{K24A/R31A}$/Vtc3/Vtc4; Vtc3$^{TM1,mutant}$, Vtc1/Vtc3$_{K694A/K698A/R705A/R709A}$/Vtc4; Vtc4$^{TM1,mutant}$, Vtc1/Vtc3/Vtc4$_{K622A/R629A}$; Vtc1$^{ΔN}$, Vtc1$_{Δ1–21}$/Vtc3 /Vtc4. Source data are provided as a Source Data file. The experiments were repeated three times independently with similar results. **b** In-cell $^{31}$P-NMR measurement of vacuolar polyP in yeast strains. *vtc3* and *vtc3$^{TM1,mutant}$*, *vtc4* and *vtc4$^{TM1,mutant}$*, *vtc1*, *vtc1$^{TM1,mutant}$* and *Vtc1$^{ΔN}$* were complemented into the *vtc2Δvtc3Δ*, *vtc2Δvtc4Δ*, *vtc2Δvtc1Δ* background strains, respectively. Chemical shifts of

vacuolar polyP are denoted. MDP was used as the std. reference. **c** smFRET analysis of VTC conformational changes. *Top*: Sites for fluorophores conjugation are highlighted in cyan on the molecular surface representation of InsP$_6$-activated VTC complex. The connection between TTM$^{Vtc4}$ and TM$^{Vtc4}$ (residues 472–618) is represented in magenta surface. The N-terminus of Vtc1$_A$ protomer (N-ter, residues 1–21) is shown in cartoon representation. *Bottom*: smFRET profiles of VTC complex (Wt) in the presence and absence of 1 mM InsP$_6$ are colored in black and gray lines, respectively. smFRET profiles of VTC complex carrying Vtc1 N-terminus deletion (Vtc1$^{ΔN}$) in the presence and absence of 1 mM InsP$_6$ are colored in blue and green lines, respectively. Red dashed line indicates the position of FRET efficiency of 0.77.

---

the polymerase domain at the entrance of the trans-membrane channel and facilitate the coupling of polyP synthesis and translocation. To fully illustrate the mechanism of VTC activation, an inactive structure in the absence of PP-InsP signal awaits to be determined.

## Discussion

In this work, the high-resolution structure of InsP$_6$-bound VTC complex provides a first snapshot of the eukaryotic polyP polymerase in its activated state. Our findings—together with a wealth of previously reported biochemical and genetic data—now offer mechanistic insights into the assembly and the coupling of polyP synthesis and membrane translocation. We resolve a partially open conformation of the trans-membrane channel in the inositol pyrophosphate-activated VTC complex, wherein the smallest radius of the channel is 1.5 Å (Supplementary Fig. 11). The narrow channel would not allow structured polyP chain passing through and ensure transport of one elongated chain at a time into the vacuolar lumen, preventing its accumulation in the cytosol where its presence as a potent metal chelator can negatively impact growth and development[46].

The VTC trans-membrane channel is about 60 Å in length (Supplementary Fig. 11), corresponding to lay a polyP chain of ~13 orthophosphate units. However, the yeast vacuolar polyP chains can be synthesized up to hundreds of units[1]. We thus speculate that the polymerization in the catalytic polymerase domain continues even after the freshly synthesized polyP chain emerges from the membrane pore. Our structure reinforces the idea that synthesis and membrane

translocation of polyP occur concomitantly, rationalizing the coupling model of polyP synthesis and membrane translocation.

The radius of a phosphate ion is 2.4 Å, thus the partially open channel observed in our structure has to undergo additional conformational changes to enable polyP membrane transport. Previous studies have reported that vacuolar polyP generation is tightly depended on the proton-gradient across membrane and the positive luminal electrical potential—energized by V-ATPase[13,28,47]. In vivo, inhibition of V-ATPase blocks VTC function[28]. Thus, we speculate that the proton-gradient and its changes in response to cell stress would regulate the operation of VTC, e.g., conformational changes in the channel itself, thereby stimulating polyP synthesis and membrane translocation. Additional structural and functional studies will be required to characterize the molecular mechanism of channel opening and closing in VTC complex.

In yeast, the activity of VTC can be promoted by an accessory subunit association, called Vtc5[30]. Although Vtc5 also harbors a SPX domain, Vtc5 appears to function independent of sensing PP-InsPs[30], consistent with our findings of the core VTC complex that only the SPX domain of Vtc4 acts as the PP-InsP sensor. Vtc5 is the only protein identified to act directly on the VTC complex to stimulate polyP production. How the regulation operated remains an open question. Biophysically, our smFRET analysis show that the movement of TTM$^{Vtc4}$ away from the trans-membrane channel is detrimentally for polyP generation. We speculate that the dwell time of TTM$^{Vtc4}$ attached at the entrance of the translocation channel may be correlated to the VTC

activity and the chain length of produced polyP. The accessory Vtc5 subunit may contribute to a regulation of this dwell time. Further total-internal reflection fluorescence (TIRF) based smFRET dynamic and kinetic studies of glass-immobilized VTC complex would provide insights into this aspect.

Two sub-complexes of Vtc1/Vtc2/Vtc4 and Vtc1/Vtc3/Vtc4 have been reported to generate yeast polyP[12,29,30]. Vtc1/Vtc3/Vtc4 complex mainly locates on vacuolar membrane, while the Vtc1/Vtc2/Vtc4 complex is found in the cell periphery and is relocalized to the vacuole under phosphate starvation[12]. The homologs of Vtc3 and Vtc2 subunits share high sequence identity (56.3%) (Supplementary Fig. 13), and the previously resolved structure of an isolated TTM$^{Vtc2}$ fragment[12] can be well modeled onto TTM$^{Vtc3}$ domain in the Vtc1/Vtc3/Vtc4 complex structure (Supplementary Fig. 14), indicating that the two sub-complexes could use similar mechanism for polyP generation.

PP-InsPs have emerged as central regulators of phosphate homeostasis in eukaryotic cells. In the case of VTC they directly control the synthesis and vacuolar transport of an important phosphate store and energy metabolite. It is of note that SPX domain receptors also occur in human and plant Pi transporters with PP-InsPs controlling of Pi efflux and import[9,48–50]. The SPX domains in these transceptors may regulate transport activity using mechanisms similar to what we described for VTC.

VTC orchestrates cellular Pi storage by generating polyP chains under sufficient Pi condition only, stimulated by high PP-InsPs levels. In many symbiotic fungi such as arbuscular mycorrhiza, the fungal VTC complex generates the cellular polyP store and transport Pi in the form of polyP to the associated plant, enabling the plant to efficiently take up Pi from the the soil[51,52]. VTC is also present in human pathogens such as trypanosomes, where the disruption of polyP synthesis can impact osmolregulation of the pathogen and thus virulence[3,53]. Our mechanistic dissection of VTC function now provides novel opportunities to exploit VTC as a target for drug design to improve plant Pi uptake from symbiotic fungi, and to treat wide-spread human diseases.

## Methods

### Manipulation and culture of yeast strains

Yeast used in this study were constructed based on a BY4741 background strain, by transformation of PCR products or plasmids containing selectable markers[54]. Details and primer sequences are listed in Supplementary Tables 2 and 3. The pRS415 plasmid was used for complementary experiments, by integrating it into corresponding yeast strains. Complemented *vtc* genes were expressed under the control of ADH promoter. SD medium (0.67% yeast nitrogen base without amino acids, supplemented with appropriate amino acids and 2% glucose) (Coolaber) was used for the screening of complemented yeast strains. Yeast strains' information are detailed in Supplementary Table 2. Constructed yeast strains were verified using PCR. Yeast cells were grown at 30 °C in YPD medium (1% yeast extract, 2% peptone, and 2% glucose).

### In-cell $^{31}$P-NMR measurement of yeast polyP

For this measurement by NMR spectroscopy, cells were cultured in YPD medium at 30 °C to a $OD_{600}$ of 0.5–0.6. 500 ml cells were harvested by centrifugation at 4 °C with 4000 × *g*. for 10 min. Harvested cells were washed and resuspended in ice-cold YPD medium at a concentration of 0.8 g ml$^{-1}$. These prepared yeasts were transferred directly to NMR tube for immediate analysis. $^{31}$P-NMR data were collected at 15 °C on a Bruker Avance III 600 MHz spectrometer equipped with a BBO probe. Spectra were acquired for about 9.5 min, by using 90° pulses at a repetition rate of 1.5 s and 250 acquisitions. 3.3 mM methylene diphosphonate (MDP, δ = 20.58 ppm) in $D_2O$ in a capillary tube was used as an identical external reference sample for each measurement and for integrals normalization. Chemical shifts were calibrated against 85% phosphoric acid (δ = 0.0 ppm). Peaks of yeast

vacuolar polyP, the terminal and central residues of polyP chain, were assigned by reference to published chemical shifts[14,35].

### Protein expression and purification

The codon-optimized DNAs of yeast *Vtc1* (Uniprot: P40046), *Vtc3* (Uniprot: Q02725) and *Vtc4* (Uniprot: P47075) were subcloned separately into the pMlink vector[55]. Vtc1 was tagged with a C-terminal HA tag, Vtc3 was tagged with an N-terminal 3×Flag, and Vtc4 was tagged with a C-terminal 2×Strep tag. The site-specific mutations were introduced into genes by overlapping PCR and were verified by DNA sequencing.

Expi293F$^{TM}$ cells (Invitrogen) were cultured in Union-293 media (Union-Biotech, Shanghai) at 37 °C under 5% $CO_2$ in a ZCZY-CS8 shaker (Zhichu Instrument). When cell density reached $2.0 × 10^6$ cells per milliliter, the cells were transiently cotransfected with three plasmids and 4 kDa linear polyethylenimine (PEI) (Polysciences). The three plasmids, ~0.67 mg of each, were premixed with 4 mg PEI in 50 ml fresh medium for 20 min before adding into 1 liter cell culture for transfection. The transfected cells were cultured for another 60 h before harvesting.

For the preparation of cryo-EM sample, the cells were collected and resuspended in the buffer containing 50 mM Tris-HCl (pH 7.4), 150 mM NaCl, 1 mM InsP$_6$, 1% LMNG (Anatrace), 0.1% CHS (Anatrace), and 0.25% Soy Phospholipids (Sigma). After incubation at 4 °C for 2 h, the solution was centrifuged at 4 °C with 56,600 × *g*. for 1 h. The supernatant was incubated with anti-Flag G1 affinity resin (GenScript) at 4 °C for 1 h, rinsed with wash (W1) buffer containing 50 mM Tris-HCl (pH 7.4), 150 mM NaCl, 1 mM InsP$_6$, 0.02% GDN (Anatrace), and eluted by W1 buffer supplemented with 250 µg ml$^{-1}$ FLAG peptide (GenScript). The eluent was then loaded to the Strep-Tactin resin (IBA) and incubated at 4 °C for 30 min, washed with wash (W2) buffer containing 100 mM Tris-HCl (pH 8.0), 150 mM NaCl, 1 mM InsP$_6$, 0.02% GDN, followed by elution with W2 buffer plus 2.5 mM Desthiobiotin (IBA). The eluent was concentrated and applied to size-exclusion chromatography (SEC, Superose 6 Increase 10/300 GL, GE Healthcare) in the buffer containing 25 mM Tris-HCl (pH 8.0), 150 mM NaCl, 1 mM InsP$_6$, and 0.02% GDN. Target fractions of VTC complex were concentrated to ~6.5 mg ml$^{-1}$ for cryo-EM grid preparation.

Sample used for the assay of polyP synthesizing activity, were purified using anti-Flag G1 affinity resin and Strep-Tactin resin in tandem. During the purification, InsP$_6$ was not added, and 0.02% DDM (Anatrace) was used instead of LMNG and GDN.

For the preparation of TTM$_{189–480}^{Vtc4}$ (residues 189–480 of Vtc4) and TTM$_{183–553}^{Vtc3}$ (residues 183–553 of Vtc3) fragments, each gene was constructed in pET-21b and pET-15d vector, respectively. Protein was expressed in *E. coli* strain BL21(DE3) using LB medium, and induced with 0.2 mM IPTG at 16 °C for 16 h. Harvested cells were lysed, and the target protein was purified over Ni$^{2+}$ affinity resin, Source 15Q and Superdex-75 Increase 10/300 columns (GE Healthcare) used in tandem. For the SEC assay, each protein was finally prepared in a buffer containing 25 mM Tris-HCl (pH 8.0), 150 mM NaCl.

### Cryo-EM grid preparation and data acquisition

3.5 µl aliquots of the purified VTC complex were placed glow-discharged holey carbon grids (Quantifoil Cu R1.2/1.3, 300 mesh). The grids were blotted with a Vitrobot Mark IV (ThemoFisher Scientific) using 3.5 s blotting time with 100% humidity at 8 °C, and plunge-frozen in liquid ethane cooled by liquid nitrogen. The cryo-grids were transferred to 300 kV Titan Krios electron microscopes (Thermo Fisher) equipped with a GIF Quantum energy filter (slit width 20 eV) and a Gatan K3 Summit detector. EPU software (v2.9) was used for fully automated data collection. Micrographs were recorded in the super-resolution mode with a magnification of 81,000×, resulting in a pixel size of 1.07 Å. Each micrograph stack, which contains 32 frames, was exposed for 3.5 s with a total electron dose of 55 e$^-$/Å$^2$. The defocus value of each image was set to −1.2 to −2.2 µm and estimated by CTFFIND4 (v4.1.14).

## Cryo-EM Data processing

A diagram of the procedures for data processing is presented in Supplementary Fig. 4. 10,033 movies were collected and motion-corrected using MotionCor2. A total of 9861 good micrographs were selected and 5,398,065 particles were automatically picked using cryoSPARC (v2.15.0) Blob picker[56] and Topaz software (v0.2.3)[57]. After several rounds of 2D classification and 3D classification, 734,934 particles were selected, followed by 3D refinement. The VTC complex yielded a particle density with an estimated resolution of 3.0 Å based on gold-standard Fourier shell correlation (FSC)[58]. Local resolution variations of the map were estimated using Reamap (v1.1.4)[59].

## Model building and refinement

The overall structure of VTC complex was firstly de nove main-chain modeled by a fully automated deep learning-based method using DeepTracer (version 1.0)[60], resulting a backbone atomic structure. Then, SPX, TTM, and TM domains predicted from AlphaFold2[61] were docked into this backbone atomic structure to aid in model building. The model was manually refined by iterative rounds of model adjusting in COOT[62]. The lipid like densities were molded with phosphatidylcholines (PC), considering the abundance of PC composition in yeast vacuolar membrane[63]. The obtained model was refined against the map using PHENIX[64] in real space with secondary structure and geometry restraints, respectively. Model quality was evaluated using the Molprobity scores[65], the Ramachandran plots, and EMRinger[66]. Figures were generated using ChimeraX (v1.2.5) and PyMol (v2.4.1).

## PAGE assay for the activity of polyP synthesis

30 μl samples containing 5 μM VTC complex, 15 mM ATP, 10 mM $MgCl_2$, 1 mM $MnCl_2$, 0.5 mM pyrophosphate (PPi), 40 mM creatine phosphate, and 40 U/ml creatine kinase were incubated with $InsP_6$ at required concentration for 4 hours at 27 °C. To favor detection of the $InsP_6$-stimulated polyP synthesis by VTC complex, a maximum of 10 mM $InsP_6$ was used in the assay. Reaction was stopped by adding 3.3 μl EDTA to a 10 mM final concentration. Then the mixture was further mixed with 6× Orange G loading buffer (10 mM Tris-HCl pH 7.0, 1 mM EDTA, 30% glycerol, 0.1% Orange G), and 20 μl sample were loaded onto a polyacrylamide slab gel (20% polyacrylamide (19:1 crylamide/bis-acrylamid), 90 mM Tris-HCl, 26 mM Boric acid, 2 mM EDTA, 0.07% APS (w/v), 0.04% TEMED) for gel electrophoresis (TBE-PAGE) and visualization. A commercial polyP with an average chain length of 60 residues (Fujifilm) was used as the marker. Assembled VTC complex used for TBE-PAGE assay have been confirmed using SDS-PAGE, as summarized in Supplementary Fig. 15.

The VTC-produced polyP were quantified using DAPI-based measurement[37]. 0.5 μl reaction mixture was mixed with 275 μl DAPI buffer (20 mM HEPES pH 7.0, 150 mM KCl) and 20 μl DAPI stock (150 μM). After incubation, 240 μl mixture was placed in a black 96-well plate and the DAPI-polyP fluorescence was measured with a EnSpire Multimode Plate Reader using $\lambda_{ex} = 415$ nm and $\lambda_{em} = 550$ nm, in a bottom-reading mode.

## Fluorescent dye conjugation and single-molecule fluorescence resonance energy transfer measurement

By substituting intrinsic cysteine of Vtc subunits (C105M in Vtc1, C106M/C751M in Vtc3, C63M/C418M/C614M/C716M in Vtc4), protein carrying specifically introduced cysteine (K415C and K689C in Vtc4) was purified in the same way as the wilt-type VTC complex, except that no $InsP_6$ was added in the purification. Alexa 488 (Thermo Fisher, A10254) and Cy5 (GE Healthcare, PA15131) were freshly dissolved in DMSO at 1 mM concentration before conjugation. Alexa 488 and Cy5 were premixed and incubated with protein sample at a molar ration of 4:4:1. The conjugation was performed for 8 h at 4 °C in dark. Excess dyes were removed by using a desalting column, and the target protein was collected for smFRET data collection.

smFRET data were collected using our MicroTime 200 system (PicoQuant) as previously described[67]. A pulsed interleaved excitation (PIE) scheme at a repetition of 32 MHz was employed for data collection to exclude the emitted photons resulted from donor only or acceptor only species[68]. Protein sample were prepared in a buffer containing 25 mM Tris-HCl (pH 8.0), 150 mM NaCl, 0.02% GDN, and 0.005% (v/v) Tween 20, with additional 1 mM ascorbic acid and 1 mM methylviologen for photobleaching and blinking minimization[69]. A concentration of about 100 pM dye-labeled protein was used for data collection. $InsP_6$ was prepared as stock solution and was mixed with dye-labeled protein to achieve the desired concentrations. The smFRET data were typically collected for about 40 min. Photon time traces were binned with a 1 ms width using SymPhoTime64 software (PicoQuant), and 6–12 counts/bin were used as the threshold for burst searching. The burst searching process was performed using our previous handwritten script[70,71], and a minimum of 35 total photon counts was defined as a burst event. The exact FRET efficiencies were calculated based on calibrated parameters for the instrumentation and fluorophores. FRET efficiency of an individual burst was calculated as $E = (F_{Dex/Aem} - LK - Di) / (\gamma \cdot F_{Dex/Dem} - F_{Dex/Aem} - LK - Di)$. $F_{Dex/Aem}$ represents the photo count for the donor excitation and acceptor emission channel, $F_{Dex/Dem}$ represents the photo count for the donor excitation and donor emission channel, LK represents the donor leakage into the acceptor channel, Di represents the acceptor emission excited directly by the donor excitation wavelength, and γ represents the detection-correction factor. Using the established methods by Shimon Weiss group[72], we have previously calibrated theses parameters for our instrumentation and fluorophores[67], with LK = 0.13, Di = 0.06, and γ = 0.47, respectively[67].

To model the fluorescent probes on our VTC structure, the fluorophores were patched onto the labeling sites using Xplor-NIH[73] (3.6), and the linker between protein backbone and rigid portion of the fluorophore were given torsion angle freedom and were allowed reorient. 1000 structures were selected for their overall energy, and the average distance between the geometric centers of the two chromophores was calculated to afford expected FRET efficiency.

## Reporting summary

Further information on research design is available in the Nature Portfolio Reporting Summary linked to this article.

## Data availability

The atomic coordinate of $InsP_6$-activated VTC complex has been deposited in the Protein Data Bank with the accession code 7YTJ. The EM map of $InsP_6$-activated VTC complex has been deposited in the Electron Microscopy Data Bank with the accession code EMD-34090. Source data are provided with this paper. Materials are available from the corresponding authors on request. Source data are provided with this paper.

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

## Acknowledgements

We thank the Cryo-EM Center, the University of Science and Technology of China (USTC), for the EM facility support. We are grateful to Dr. Yongxiang Gao (USTC) for technical support during EM image acquisition. We thank professor Haining Du and professor Xiangdong Gao, Wuhan University, for help and suggestion in yeast manipulation. We thank the Center for Protein Research, Public Laboratory of Electron Microscopy and Physical and Chemical Analysis Center, Huazhong Agricultural University, for technical support. This work was supported by the National Key R&D Program of China (2018YFA0507700 to Z.L. and P.Y.), the National Natural Science Foundation of China (32071226 to Z.L.), the Foundation of Hubei Hongshan Laboratory (2021HSZD011 and 2021HSZD016 to Z.L., J.Y., and P.Y.), and the HZAU-AGIS Cooperation Fund (SZYJY2022022 to Z.L.). Z.G. acknowledges the support of National Postdoctoral Program for Innovative Talents (BX2021108).

## Author contributions

Z.L. conceived and supervised the project. Z.G., J.C., R.L., and Y.C. designed all experiments. J.C. prepared cryo-EM sample. Z.G. and Q.X. collected EM data and determined the structure. Y.C. collected and analyzed NMR data. R.L., Z.D., and M.C. performed biochemical, cellular, and smFRET experiments. J.H., W.Z., W.M., B.W., Q.W., J.Z., P.C., H.C., J.C., D.Z., J.Y., and P.Y. contributed to data analysis and manuscript preparation. M.H. and Z.L. wrote the manuscript.

## Competing interests

The authors declare no competing interests.
