## [Peer Review File · Nature Communications]

The cytoplasmic synthesis and coupled membrane translocation of eukaryotic polyphosphate by signal-activated VTC complexREVIEWER COMMENTS

Reviewer #1 (Remarks to the Author):

This work proposed a high-resolution structure of InsP6-bound VTC complex, which is important to understand how the eukaryotic polyP polymerase activity is regulated. The detailed structure of the Vtc1/3/4 complex includes: 1) the TM region contains 15 TM helices (3 subunits of TM-Vtc1; 1 TM-Vtc3 and TM-Vtc4); 2) Position of SPX and TTM of Vtc3 and Vtc4. By introducing different mutations in SPX of Vtc3 and Vtc4, they demonstrated that it is the SPX of Vtc4 rather than SPX of Vtc3 sensed PP-InSP and stimulate the polyP synthesis. In addition, they also found that the TM domains of Vtc1, Vtc3 and Vtc4 constitute TM region of the complex and determines the polyP synthesis in yeast. The work is important not only for understanding the regulatory mechanism by which the yeast polyP synthesis is control, but also will be important reference for understanding of the molecular regulation of SPX-domain containing proteins in plants and human. Overall, it is important data that should be published.

Minor comments:

- 1) Figure 1a is not an original data. Should be moved to supplemental materials.
- 2) Supplementary Fig. 1 showed the functions of each members in the Vtc complex. The data should be either in result (if it is novel) or removed. The citation of the Fig (line 71) is not appropriate.
- 3) The speculation in Lines 145-147 has no support data. Need either more explanation or be removed.
- 4) Lots of data interpretation were written in the legend of the Figures or supplemental Figures. Please add more interpretations in the results.

Reviewer #2 (Remarks to the Author):

The manuscript by Guan et al. is well-written and technically sound. It provides valuable insights regarding a fundamentally interesting and intriguing molecular machine - the vacuolar transporter chaperone complex. The proposed structural model and mechanistic analysis of polyphosphate transport appear reliable and consistent with the presented data. I have no concerns and the manuscript is acceptable as is.

Reviewer #3 (Remarks to the Author):

In the manuscript by Guan et al., the authors present a Cryo-EM structure of the activated VTC complex at 3.0 Å resolution, which is responsible for the synthesis and translocation of inorganic polyphosphate (polyP). PolyP synthesis is regulated by inositol pyrophosphate (PP-InSP), which is directly sensed by the complex. The complex consists of 3 subunits (Vtc1, Vtc3, and Vtc4), which assemble in a 3:1:1 stoichiometry. Overall this is a highly interesting and well-resolved structure, which gives mechanistic hints about its function.

But there are numerous points that need clarification and improvement:

1. Show and reference figures in the order they appear in the text and make sure they fit each other (e.g. line 71 – refer to supplementary figure 1, but S1 is on 31P-NMR measurements, ...)
2. Supplementary figure 2d: dose-dependent activity assay. It seems that at lower concentrations of InsP6 (e.g. 1mM), longer polyP chains are produced compared to 10 mM. Can the authors comment on this and is a quantification of the activity possible?
3. Supplementary figure 5b: Position of InsP6 bound to SPX (of Vtc4 in the complex) seems to be slightly different compared to the previously reported isolated SPX (of Vtc4) structure. Can the authors compare the coordination geometry of both complexes and highlight commonalities and differences?
4. The In-cell 31P-NMR measurements are suddenly mentioned in line 190, but this

assay has not been really introduced and is only mentioned briefly. This is an elegant method to monitor function and would deserve a better introduction and description in the text. It is a powerful in-vivo assay and places the structure nicely in context.

5. What is the experimental evidence that the produced polyP in the in vitro assays passes through the channel?

6. The single-molecule FRET data are at this stage not convincing and I would recommend removing this part. Single-molecule FRET is a powerful tool to monitor conformational transitions, but also requires many more controls, which are absent in the current manuscript.

7. Stick to one-letter or three-letter code for amino acids.

8. Please show a figure with the channel path (e.g. surface representation with the molecule cut in half to illustrate the channel incl. properties; estimate the diameter of the channel; without phosphate and the lipids) – would you expect the channel to be open all the time or is channel opening and closing regulated?

9. Currently the discussion section mainly focuses on general aspects of VTCs and polyP but does not really place the obtained structure in context. Although I agree that a single structure cannot explain all mechanistic details, it allows speculating how different parts of the machinery could function. Here are some points one could touch on or speculate:

- How do the 2 VTC complexes (Vtc1/Vtc2/Vtc4 versus Vtc1/Vtc3/Vtc4) differ?

- How to start the reaction? Is a polyP Primer needed? What determines the length of polyP? (dependent on channel length?) What is the conformation of polyP in the channel (elongated, structured?)

- The coupling of polyP synthesis to translocation is not entirely clear to me – is an opening of the channel required? What terminates the reaction?

10. Why did the authors use the VTC complex purified in DDM for functional assays and not in LMNG or GDN as used for structural work? Is the activity different in these detergents?

11. Language and spelling in particular in the 'Model building and refinement' section in Material and Methods needs to be improved.

REVIEWER COMMENTS

Reviewer #1 (Remarks to the Author):

This work proposed a high-resolution structure of InsP6-bound VTC complex, which is important to understand how the eukaryotic polyP polymerase activity is regulated. The detailed structure of the Vtc1/3/4 complex includes: 1) the TM region contains 15 TM helices (3 subunits of TM-Vtc1; 1 TM-Vtc3 and TM-Vtc4); 2) Position of SPX and TTM of Vtc3 and Vtc4. By introducing different mutations in SPX of Vtc3 and Vtc4, they demonstrated that it is the SPX of Vtc4 rather than SPX of Vtc3 sensed PP-InSP and stimulate the polyP synthesis. In addition, they also found that the TM domains of Vtc1, Vtc3 and Vtc4 constitute TM region of the complex and determines the polyP synthesis in yeast. The work is important not only for understanding the regulatory mechanism by which the yeast polyP synthesis is control, but also will be important reference for understanding of the molecular regulation of SPX-domain containing proteins in plants and human. Overall, it is important data that should be published.

Response: Thank you very much for the positive comments about insights from our study and for the constructive suggestions that have helped us to improve our manuscript. According to your suggestions, we have carefully revised our manuscript. Please find our point-by-point response to your suggestions listed below.

Minor comments:

1) Figure 1a is not an original data. Should be moved to supplemental materials.

Response: We have moved Figure 1a to Supplementary Figure 1 in our revised Supplementary Information.

2) Supplementary Fig. 1 showed the functions of each members in the Vtc complex. The data should be either in result (if it is novel) or removed. The citation of the Fig (line 71) is not appropriate.

Response: Thank you very much for this constructive suggestion. To our knowledge, it is the first time to use in-cell ³¹P-NMR measurements to characterize the functions

of each Vtc members in BY4741 yeast strain. In our revised manuscript, we have moved these results into the main text (paragraph 1 in page 5) and now present this figure as main Figure 1. The citation has been removed, as suggested.

3) The speculation in Lines 145-147 has no support data. Need either more explanation or be removed.

Response: Thank you for pointing this out. Our revised statement (Lines 196-198) "The possible physiological role of our observed InsP_6 -promoted TTM^{Vtc4} - TTM^{Vtc3} interaction needs further exploration."

4) Lots of data interpretation were written in the legend of the Figures or supplemental Figures. Please add more interpretations in the results.

Response: Thank you for this constructive suggestion. In the revised manuscript, we have moved these statements into the main text and included additional explanations, including the in-cell ^{31}P -NMR measurements of polyP in yeast strains (paragraph 1 in page 5), the structural characterization of small molecules (paragraph 2 in page 7), and the InsP_6 -promoted TTM^{Vtc4} - TTM^{Vtc3} interaction (paragraph 2 in page 9).

Reviewer #2 (Remarks to the Author):

The manuscript by Guan et al. is well-written and technically sound. It provides valuable insights regarding a fundamentally interesting and intriguing molecular machine - the vacuolar transporter chaperone complex. The proposed structural model and mechanistic analysis of polyphosphate transport appear reliable and consistent with the presented data. I have no concerns and the manuscript is acceptable as is.

Response: Thank you very much for the positive comments.

Reviewer #3 (Remarks to the Author):

In the manuscript by Guan et al., the authors present a Cryo-EM structure of the activated VTC complex at 3.0 Å resolution, which is responsible for the synthesis and translocation of inorganic polyphosphate (polyP). PolyP synthesis is regulated by inositol pyrophosphate (PP-InsP), which is directly sensed by the complex. The complex consists of 3 subunits (Vtc1, Vtc3, and Vtc4), which assemble in a 3:1:1 stoichiometry. Overall this is a highly interesting and well-resolved structure, which gives mechanistic hints about its function.

But there are numerous points that need clarification and improvement:

Response: Thank you very much for the positive comments about insights from our study and for the very constructive suggestions that have helped us to improve our manuscript. According to your suggestions, we further performed additional experiments and analyses. We have carefully revised our manuscript. Please find our point-by-point response to your comments listed below.

1. Show and reference figures in the order they appear in the text and make sure they fit each other (e.g. line 71 – refer to supplementary figure 1, but S1 is on ³¹P-NMR measurements, ...)

Response: Thank you for pointing this out. We have removed this inappropriate citation, and have carefully proofread our manuscript. All figures now appear in the correct order and cited correctly in the main text.

2. Supplementary figure 2d: dose-dependent activity assay. It seems that at lower concentrations of InsP₆ (e.g. 1 mM), longer polyP chains are produced compared to 10 mM. Can the authors comment on this and is a quantification of the activity possible?

Response: Thank you for this suggestion. In revision we re-performed the InsP₆ dose-dependent activity assay and used DAPI-based measurement to quantify the activity (Response Fig. 1). The increasing trend of polyP levels is consistent with the TBE-PAGE visualization. We have included these quantification data and added a figure as Supplementary Fig. 2d. Regarding to your question about some longer polyP

chains produced at lower InsP_6 concentrations, longer chains were also observed in shorter reaction times (e.g. 2 hours) compared to 10-12 hours. This may be because the polyP produced at higher concentrations (e.g. 10-12 hours reaction time, or 10 mM InsP_6 stimuli) are less stable and undergo some degradation, which converts them into short chains.

Response Fig. 1. TBE-PAGE visualization of polyP synthesized by Vtc1/Vtc3/Vtc4 complex at different InsP_6 concentration (left), and quantification of the produced polyP by using DAPI-based measurement. The reactions were performed at required InsP_6 concentrations. 5 μM protein complex were used and reactions were performed for 4 hours.

3. Supplementary figure 5b: Position of InsP_6 bound to SPX (of Vtc4 in the complex) seems to be slightly different compared to the previously reported isolated SPX (of Vtc4) structure. Can the authors compare the coordination geometry of both complexes and highlight commonalities and differences?

Response: InsP_6 ligand binds to the basic surface patch of SPX proteins, but it is not very well coordinated. Conformation differences of InsP_6 in some reported SPX/ InsP_6 complex structures have been observed before. In the revised manuscript, we have compared these complexes, added a figure as Supplementary Fig. 7, and included a discussion in the main text (lines 143-147).

4. The In-cell ^{31}P -NMR measurements are suddenly mentioned in line 190, but this

assay has not been really introduced and is only mentioned briefly. This is an elegant method to monitor function and would deserve a better introduction and description in the text. It is a powerful in-vivo assay and places the structure nicely in context.

Response: Thank you very much for this constructive suggestion. In our revised manuscript, we have added an introduction in the main text (paragraph 1 in page 5) and presented our previous Supplementary Fig. 1 as main Figure 1, to highlight the powerful in-vivo assay.

5. What is the experimental evidence that the produced polyP in the in vitro assays passes through the channel?

Response: To assess whether the VTC-synthesized polyP in vitro passes through the trans-membrane channel of VTC complex, we subjected the reaction mixture to size-exclusion chromatography (SEC). We collected the VTC complex fractions and used DAPI (4',6-diamidino-2-phenylindole) (*Journal of Fluorescence*, 2008, 18(5):859-66), a polyP-detecting dye, to probe if the produced polyP is retained in the VTC complex. We found that significant polyP-DAPI signal can be detected for the collected SEC fractions of the VTC complex, indicating that the produced polyP associates with the VTC complex and may be transited into the trans-membrane channel. We have included these data in the revised manuscript (paragraph 2 in page 5) and added figures as Supplementary Fig. 2e, f. Additional follow-up work, more direct evidence for polyP passing through the channel requires further determining the complex structure of VTC and produced polyP.

6. The single-molecule FRET data are at this stage not convincing and I would recommend removing this part. Single-molecule FRET is a powerful tool to monitor conformational transitions, but also requires many more controls, which are absent in the current manuscript.

Response: We completely agree with you that smFRET is a powerful tool to monitor conformational transitions. smFRET is also a robust method to characterize

ligand-induced conformational changes in protein. In our study, we apply smFRET to monitor the movement between TTM^{Vtc4} and trans-membrane channel following the InsP₆ sensing. By this movement, PP-InsP binding orients the catalytic polymerase domain at the entrance of the trans-membrane channel. These results helped us to understand the mechanism of VTC activation and the coupling of polyP synthesis and membrane translocation. Moreover, smFRET results helped us to define the role of Vtc1 N-terminus in TTM^{Vtc4} – TM region interaction. We thus would like to include smFRET data in our revised manuscript. We have however performed additional experiments and revised the description of these data:

Firstly, additional new experiments showed that the fluorescent probes have been specifically conjugated. Furthermore, this probes-conjugated VTC complex exhibits polyP-synthesizing capacity comparable to wild-type VTC complex, and also in an InsP₆ dose-dependent manner. These results indicate that probes do not perturb VTC structure and function. We have included these results in the main text (lines 261-270), and added figures as Supplementary Fig. 12a, b.

Secondly, the exact FRET efficiencies were calculated based on calibrated parameters for the instrumentation and fluorophores. We apologize for missing some details in smFRET data analysis. FRET efficiency of an individual burst was calculated as $E = (F_{\text{Dex/Aem}} - \text{LK} - \text{Di}) / (\gamma \cdot F_{\text{Dex/Dem}} - F_{\text{Dex/Aem}} - \text{LK} - \text{Di})$, in which $F_{\text{Dex/Aem}}$ represents the photo count for the donor excitation and acceptor emission channel, $F_{\text{Dex/Dem}}$ represents the photo count for the donor excitation and donor emission channel, LK represents the donor leakage into the acceptor channel, Di represents the acceptor emission excited directly by the donor excitation wavelength, and γ represents the detection-correction factor. Using the established methods by Shimon Weiss group (*Biophys J.* 2005, 88(4):2939-2953), we have previously calibrated these parameters for our instrumentation and fluorophores, with LK = 0.13, Di = 0.06 and $\gamma = 0.47$, respectively (*Biomolecules.* 2021, 11(9):1321). We have included these details in the revised Methods

Thirdly, the experimental observed FRET efficiency of the InsP₆-enriched high-FRET species is consistent with the theoretical calculation. Based on our InsP₆-activated VTC complex structure, we modeled the fluorescent probes at their labeling sites and calculated the probe-probe distance. The calculated average distance is $42.6 \pm 6.5 \text{ \AA}$ between geometric centers of the two chromophores, expecting a theoretical FRET efficiency of 0.77. This value is consistent with the observed efficiency of the InsP₆-enriched high-FRET species. This comparison and consistency both corroborate our InsP₆-activated VTC structure and smFRET data. We have included this analysis in the main text (lines 276-282), and have added a figure as Supplementary Fig. 12c. To model probes and calculate the position distribution of the conjugated fluorophores on the VTC complex structure, we used Xplor-NIH (*Protein Science*. 2018, 27(1):26-40) to perform Monte Carlo simulations. These simulations could account the orientation freedom and geometric accessible volume of chromophores. Details have been included in the revised Methods. We have previously used this method to calculate theoretical FRET efficiency in the experimental structures of other protein, and successfully corroborated that the theoretical values are consistent with that of experimentally measured (*Cell Discovery*. 2019, 5:19; *Cell Discovery*. 2019, 5:29).

7. Stick to one-letter or three-letter code for amino acids.

Response: Thank you for pointing this out. All amino acids in the manuscript are coded using one-letter now.

8. Please show a figure with the channel path (e.g. surface representation with the molecule cut in half to illustrate the channel incl. properties; estimate the diameter of the channel; without phosphate and the lipids) – would you expect the channel to be open all the time or is channel opening and closing regulated?

Response: Thank you for the insightful comment and constructive suggestion. We have analyzed the channel path and added a figure as Supplementary Fig. 11. The narrowest position of the channel is below the observed phosphate and is surrounded by R31 of TM1^{Vtc1}, R709 of TM1^{Vtc3}, and R629 of TM1^{Vtc4}, with a radius of 1.5 Å.

This value is smaller than the size of a phosphate ion (radius of 2.4 Å). We therefore speculate that we captured a partially open channel conformation in the absence of produced polyP. We have included this analysis in the revised manuscript (lines 224-226), and proposed possible regulations in Discussion (paragraph 2 in page 14, page 15, and paragraph 1 in page 16).

9. Currently the discussion section mainly focuses on general aspects of VTCs and polyP but does not really place the obtained structure in context. Although I agree that a single structure cannot explain all mechanistic details, it allows speculating how different parts of the machinery could function. Here are some points one could touch on or speculate:

Response: Thank you very much for this constructive suggestion. We have discussed more in the revised manuscript, as listed below:

- How do the 2 VTC complexes (Vtc1/Vtc2/Vtc4 versus Vtc1/Vtc3/Vtc4) differ?

Response: We speculate that the two sub-complexes would use similar mechanism to generate polyP. We have added a discussion in the revised manuscript (paragraph 2 in page 16), and added figures as Supplementary Figs. 13 and 14.

- How to start the reaction? Is a polyP Primer needed? What determines the length of polyP? (dependent on channel length?) What is the conformation of polyP in the channel (elongated, structured?)

Response: Our previous studies found that the pyrophosphate acts as a primer to start the synthesis of polyP chain (*Science*. 2009, 324(5926):513-516). We have included this discussion in the revised manuscript (lines 153-155).

Based on our studies, we speculate that one elongated chain is allowed to transport at a time (paragraph 2 in page 14), and the polymerization in the catalytic polymerase domain continues even after the freshly synthesized polyP chain emerges from the membrane pore (paragraph 3 in page 14 and paragraph 1 in page 15).

- The coupling of polyP synthesis to translocation is not entirely clear to me – is an opening of the channel required? What terminates the reaction?

Response: In the revised manuscript, we discussed the contributions of proton-gradient across vacuolar membrane and the accessory Vtc5 subunit to the regulation of VTC complex. We speculate that the proton-gradient and its changes in response to cell stress would regulate the operation of VTC, e.g. conformational changes in the channel itself, thereby stimulating polyP synthesis and membrane translocation (paragraph 2 in page 15). We propose that the dwell time of TTM^{Vtc4} attached at the entrance of the translocation channel may be correlated to the VTC activity and the chain length of produced polyP. The accessory Vtc5 subunit may contribute to a regulation of this dwell time (paragraph 3 in page 15 and paragraph 1 in page 16). However, future studies are needed.

10. Why did the authors use the VTC complex purified in DDM for functional assays and not in LMNG or GDN as used for structural work? Is the activity different in these detergents?

Response: We prepared VTC complex in 0.02% GDN, which enabled us to get high-quality cryo-EM micrographs for structure determination. In 0.02% DDM, we found that the VTC complex exhibits similar polyP-synthesizing activity as in 0.02% GDN (Response Fig. 2).

Response Fig. 2. The VTC complex exhibits similar polyP-synthesizing activity in

DDM and GDN. The reactions were performed at required InsP₆ concentrations. 5 μM protein complex were used and reactions were performed for 4 hours. The produced polyP were quantified by using DAPI-based measurement.

11. Language and spelling in particular in the ‘Model building and refinement’ section in Material and Methods needs to be improved.

Response: We are sorry for this in our previous manuscript. We have corrected them and carefully proofread the revised manuscript.

REVIEWERS' COMMENTS

Reviewer #1 (Remarks to the Author):

The revised manuscript has responded the reviewer's comments effectively. I have no further comment.

Reviewer #3 (Remarks to the Author):

I am happy with the way the authors have addressed the comments and suggestions in the revised version of the manuscript. Additional information and experiments have been added. I recommend the publication of the manuscript in its current form.

REVIEWER COMMENTS

Reviewer #1 (Remarks to the Author):

The revised manuscript has responded the reviewer's comments effectively. I have no further comment.

Response: We are delighted that your comments have been effectively responded. Thank you very much for your constructive suggestions that have helped us to improve our manuscript.

Reviewer #3 (Remarks to the Author):

I am happy with the way the authors have addressed the comments and suggestions in the revised version of the manuscript. Additional information and experiments have been added. I recommend the publication of the manuscript in its current form.

Response: We are delighted that your comments have been addressed. Thank you very much for your constructive suggestions that have helped us to improve our manuscript.